# Beyond Choice: Reading Sigmund Freud at the End of *Roe*

## Karen McFadyen

Department of English and Film Studies, University of Alberta, Edmonton, AB T6G 2R3, Canada; kpmcfady@ualberta.ca

**Abstract:** After *Roe v. Wade* was overturned by the Supreme Court, pregnant people lost their Constitutional protection of abortion. The new, visible politics of susceptibility have invited a revisitation to the psychoanalytic work of Sigmund Freud. This article examines the trauma narrative of Freud's *Beyond the Pleasure Principle* and the theory of the death drive in elaborating the enduring cultural investment in protecting fetal life while examining its implications for pregnant subjects.

**Keywords:** abortion; psychoanalysis; Sigmund Freud; death drive; trauma; susceptibility; David Cronenberg



## 1. Introduction

After the Supreme Court of the United States overturned *Roe v. Wade*, ending Constitutional protection to abortion access before fetal viability, psychoanalytic scholarship had to reckon with what had been underexplored territory within its discipline. The silence around abortion was striking enough in 1969 for Hilda Abraham to address it, attributing it to an "active wish among psychoanalysts not to think about it" [1] (154). After fifty years, there is a different resonance to this apprehension: not of whether to speak of abortion, but of how to address it. For feminist psychoanalyst Naomi Snider, among others, the proximity of pro-choice rhetoric to neo-liberal values of personal choice, individual rights and radical autonomy has made for a troubled sense that the politics of reproductive rights deny the complexity of living within an oppressive social matrix [2]. As abortion is a lived experience that is so exigently political, the abortion debate does not easily reconcile itself with theoretical elaboration. The feminist impulse to know abortion in its full weight of historical and intersectional complexity is impinged upon by the stark, unyielding reality of its legal and bodily consequences. Now that the notion of choice being rhetorically compromised has become actualized through the United States Supreme Court and the susceptibility of potentially childbearing subjects has renewed visibility, the question of how abortion is disciplined, in both the legal and academic sense, has assumed a renewed and vital urgency. The interrelations of gender and otherness essential to Freud's construction of the death drive are at the center of the disciplinary impulses concerning the excessive and troubling subject of abortion. In this article, I will explore the implications of this particular disciplining framework through a narratological analysis of three intersecting objects: Sigmund Freud's retelling of a scene from *Gerusalemme Liberata* in *Beyond the Pleasure Principle*, the deciding opinion that ended *Roe* and the allegorical presence of abortion in David Cronenberg's body horror film *Crimes of the Future*.

## 2. Beyond the Pleasure Principle: Gender and the Death Drive

*Beyond the Pleasure Principle* was Freud's speculation on trauma. Amidst the uncertainty and destruction of the First World War, Freud witnessed a unique relationship to time and movement among his patients that did not accord with the prevailing theory of the pleasure principle. While action under the pleasure principle seeks to alleviate tension, a death-driven subject—a shell-shocked veteran, a helpless and confused child—will repeat actions without purpose or satisfaction. In fact, the repeated actions will be directly tied

to the scene of trauma. For Freud, this suggested that there was a pivotal return to which the unconscious is oriented: a state of death and quiescence that precedes the forward momentum of life. Alternatively, in Freud's own words:

"*It seems, then, that an instinct is an urge in organic life to restore to an earlier state of things* which the living entity has been obliged to abandon under the pressure of external disturbing forces; that is, it is a kind of organic elasticity, or, to put it another way, the expression of the inertia inherent in organic life" [3] (p. 36)

The death drive produces a sense of narrative structure within life, as Peter Brooks would explore in "Freud's Masterplot". Here, the death drive confirms that death—or, in a narratological framework, the ending—is in a dynamic relation to the beginning, a vital constitutive force in the creation of a narrative. These resurgent and ubiquitous aspects of narrative—tropes, genre conventions, motifs—are in the service of mastering the tensions of an "inescapable middle [suggestive] of the daemonic" [4] (288). The plot consists of detours and deviances on the course to resolution: the restoration of order and certainty that becomes a satisfying ending.

For Freud, the death drive emerges in a narrative as a limit of one's own capacity to inscribe an ending. In *Beyond the Pleasure Principle*, he illustrated the death drive by re-narrating a pivotal scene from Torquato Tasso's epic romance *Gerusalemme Liberata.* Here, the death drive is an unconscious orientation that mobilizes the hero into an action that becomes an unwilling repetition of a traumatic scene. Again, it is through Freud's own words that the story settles into the contours of the drive:

Its hero, Tancred, unwittingly kills his beloved Clorinda in a duel while she is disguised in the armour of an enemy knight. After her burial he makes his way into a strange magic forest which strikes the Crusaders' army with terror. He slashes with his sword at a tall tree; but blood streams from the cut and the voice of Clorinda, whose soul is imprisoned in the tree, is heard complaining that he has wounded his beloved once again. [3] (p. 22)

This scene proves that passivity is critical to the experience of the death drive, for it is only unconsciously that Tancred twice administers a fatal wound to Clorinda. He is driven by compulsion, not desire or pleasure; his wounds are fateful rather than fatal. This mythic scene liberates Freud's speculative voice, as it is not in the interest of scientific legitimacy but of the value of readerly identification that Tancred becomes a worthy case study for a nascent concept. The unconscious makes no distinctions between fantasy and reality, after all, so the very existence of this story and of a recognition of Tancred's plight is sufficient for it to have a real impact.

This does, however, rely upon a reader who, like Freud, is primed to align themselves with Tancred. While the excerpt is effective in its concision, its brief allusion to Tancred's alignments in the Crusades reduces the historical realities intersecting within this story to a suggestion. Tancred is a white Christian crusader who has repeatedly inflicted violence against Clorinda, a Muslim warrior woman: a fact that Cathy Caruth, who utilized Freud's account in her theory of trauma, titled "The Wound and the Voice", was made to account for in revisiting responses to her work:

the annihilation of experience at the core of what we think of as personal trauma is never wholly extricable from larger social and political modes of denial [. . .] the poem does not simply represent the conflict between individual or collective stories but also dramatizes their simultaneous severing and binding together, in a history that can only be figured as a speaking wound [5] (p.121)

Despite the salience of Caruth's argument, it does little to disabuse this prevailing reading of Tancred and Clorinda as unified in this figurative representation of trauma. While Tancred relies on Clorinda's presence in order to assume the sequence of the identity positions—crusader, lover, bereaved—that become his character in the epic cycle, Clorinda's presence is at this point confined to the staid inertia of the tree. Tancred's trauma relies on

his misrecognition of Clorinda: on his inability to situate her, primarily, as a woman (for his first fatal encounter with her occurred while she was disguised as a male soldier) until she has been wounded and cries. Tancred defines the limits of subjectivity: not Clorinda, whose own perspective is denied in Freud's trauma scene. She invites and refuses the reader's speculation; does she recognize Tancred as he approaches her? Is she imagining a reunion with her lover? Does she experience a surge of erotic possibility before becoming an emblem of the stultification of the death drive? We have no certainty about Clorinda, because the death drive is primed on identification with the agential body of Tancred so that we recognize the false edifice of his certainty of himself as a subject. In other words, Tancred's failure is that he does not recognize that he is driven by unconscious impulses: that he is not the master of his own fate. Clorinda has no pretense of mastery; her role is to unsettle, for her death to give an imperative and direction to Tancred's drive.

Ultimately, the sense of mastery and finality conferred by the drive fails in its encounter with otherness, particularly the otherness of a gendered body subtended by erotic and violent desire. For Tancred and Clorinda, the end is not simply delayed; it is perpetually re-encountered, caused by violence and spurred by the same death, many times over. At the intersections of gender, failed interpretation and recursive endings is Freud's affirmation of a drive so imperturbable and inescapable that it becomes fantasy, the motivating plot of an epic romance. If a narrative does not simply address the death drive, as Brooks claims, but is formed by it in impulses toward closure, the question of mastery and failure as it coalesces around these gendered objects takes on a renewed urgency in the post-*Roe* era. Without *Roe*, childbearing bodies in America have no protected access to safe abortion. This does not mean an end of death, as abortion restrictions will lead to a significant increase in maternal mortality rates [6], but within a death-driven framework, the ambivalence of an ending is pertinent. Here, the problem of abortion is, in part, a problem of narrative and the faculties of interpretation resting in the legal institution whose primacy is in their alignment with interpreting the United States Constitution: the original document of citizen identity.

### 3. Dobbs v. Jackson Women's Health Organization: The Supreme Court Makes Abortion History

The decision to overturn *Roe v. Wade* was made on 24 June 2022. Somewhat ironically, this decision was delivered prematurely; the draft opinion of Justice Samuel Alito was leaked by an anonymous source to the press in early May, sparking nationwide protest. Alito, for his part, expressed his fear that he and his fellow conservative justices could become assassination targets as a consequence of the leak. These outraged citizens, he said, could be made to believe that "they could prevent [the end of *Roe*] from happening by killing one of us" [7]. Why should Alito imagine such a deadly consequence for the revelation of a decision six weeks in advance of its final ruling?

The answer to this, I believe, is a matter of interpretation. Namely, in Samuel Alito's privileged stature as one of nine sanctioned interpreters of the Constitution, he stakes his sense of life in the writ of the United States Constitution. The *Dobbs* ruling that ended *Roe* was founded on a redress of a vital error in the original document; that *Roe* was, itself, the work of an "egregiously wrong" misinterpretation that was "on a collision course with the Constitution from the day it was decided" [8] (p. 5). Overturning *Roe* was a matter of proving, to some degree of rhetorical certainty, that the Constitution in its original intent prohibited abortion as a federally protected right. This necessitated, among other things, a return to the common law of the time. To make the Constitution right would be to restore the nation to a condition contemporary with the founding fathers, or, in Alito's words,

> The inescapable conclusion is that a right to abortion is not deeply rooted in the Nation's history and traditions. On the contrary, an unbroken tradition of prohibiting abortion on pain of criminal punishment persisted from the earliest days of the common law until 1973. The Court in Roe could have said of abortion exactly what Glucksberg said of assisted suicide: 'Attitudes toward [abortion]

have changed since Bracton, but our laws have consistently condemned, and continue to prohibit, [that practice]. (substitutions original) [8] (p. 25)

Abortion is negated through its silence. The Constitution, the foundation of a modern nation, underwrites the United States from its emergence as an independent country to its establishment as a global superpower in the modern and post-modern era, becoming a dead document at the very moment that it encounters the living impulses of otherness. The Constitution has made history and become history. Rather than revealing the textures of life in its processes of being read and interpreted, the work of interpretation that Fredric Jameson identifies as the vitality of history for those who live by and through it [9], the Constitution is captured in stasis. It is the unconscious not of the nation but of its long-dead architects, products of a world that was only emergent in the time of its creation, and has been a work in progress from ratification to present. The Constitution, for Alito, is as alive as it needs to be. Within the sanctity of the Supreme Court chambers, it is captured within a precise discursive limit—the five conservative justices who have, over decades, staked a significant portion of political capital on overturning *Roe v. Wade*.

In a 2014 interview with the conservative magazine *American Spectator*, Alito produced the iPad he used to work on his briefs [10]. While the goal there was arguably to refute the notion that a then-sixty-something Bush appointee was completely out of step with the modern age, it produced a notion that Alito is the flesh and voice through which the Constitution intersects between a living present and the inert passivity of history. It is not simply that the Constitution can be brought up onto any interface—as any remotely tech-savvy individual could make the same thing happen—but the fact that it is Alito's interface, his flesh that marks the document, and that the life of the Constitution becomes conditional upon his interpretive interest. Within the justice's chamber, the writ of dead men is life-affirming for Alito. Within a public who encounters the *Dobbs* decision through its own interfaces and communication networks, interpretation falls to the volatility of a collective readership. While *Dobbs* disciplines abortion, it fails to discipline the practice of reading. The stasis of the Constitution is vulnerable to the ravenous eyes of a critical public, and fleshly proxies are struck by the death anxiety of their identification. If the Constitution is dead, then in the eyes and minds of others, so are its most quiet and contained readers. As the excessive signifier of abortion refuses to be disciplined, the very sense of interpretive power is cast into the studied relief of susceptibility.

## 4. Terminal Fantasy: The Problem of Abortion within Narrative

What, therefore, is abortion within this narrative framework? Current psychoanalytic scholarship, reckoning with this paradigm of personal choice and autonomy against the legacies of reproductive genocide in eugenics campaigns and the forced sterilizations of Black and Indigenous people, have subtended abortion as the "generative potential" of liberation from undesired pregnancy [11] (p. 1) and the troubling conditions of being a desiring subject who is built, after a fashion, to host multiple subjects at once [12]. Ultimately, the social value of the fetus rests in its representation of endlessly deferred potential. As long as it is not actualized, wrote Katie Gentile, the fetus can be the perfect vessel through which we might save the future without actually mobilizing ourselves against the more imminent threat of climate crisis. As Gentile argues,

"the future can be saved without actually impacting the environment at large. Ambivalence can be acted out without resolution. The future can materialize not in a body that is of the world, but one that is yet to be. Using the fetal body enables a tight, circular, compulsive repetition of a present that never has to unfold into a future, since once born, the fetus ceases to be a fetus. But as with all such compulsive repetitions, anxiety is not quelled with rescuing, but instead is maintained or fueled." [13] (p. 7)

In other words, the maintenance of the fetus, as a fetish, is the real manifestation of the death drive. It is a site of endless repetition because it is necessarily unactualizable;

each fetus, once born, loses its symbolic power to a culture that desires a future that is against its destructive impulses. Much like the cry of Clorinda that alerts Tancred to her presence, the cry of a baby born alive is a moment of unbearable recognition. The drive mobilizes the forces of fetal protection back to the fences around clinics, bearing images of dismembered fetuses and holding vigils in the name of the unborn. The cry of a born baby is a demanding cry of urgent need; the silence of the fetus, as highlighted in Bernard Nathanson's impactful pro-life film *The Silent Scream*, is a call that requires no resolution or redress. It is, as Gentile has written, endlessly deferred possibility.

An abortion, therefore, is utter termination. It is not a narrative ending that emerges in dynamic interrelation with the beginning but is a collapse of the two into an unfulfilling and abject refusal. As Brooks wrote, "we detect some illumination of the necessary distance between beginning and end, the drives which connect them but which prevent the one collapsing back into the other" [4] (p. 295). In this narrative sense, desire motivates the beginning, and death—quiescence—is its end. A framework of abortion as choice would hold that abortion is terminal desire and that birth is desire shifted from the childbearing body to the child body, its imminent and urgent needs becoming the orienting desires in the lives emergent from the birth mother and child. The aborting person, as Beck has pointed out, has no name; they are unidentified as if through the silence and quiescence of the unborn, except with the punishing appellation of "murderer" cast—the cry of demand that supplants the cry of the needing baby [11] (p. 9). In a libidinal framework, as selectively deployed by Jamieson Webster and Naomi Snider, "choice" is suggestive of desire. It is, I believe, imperative to recognize that abortion is not always an outcome of choice: that the body, in its totality, is not completely subordinate to individual consciousness. Spontaneous abortions, abortions obtained out of necessity for an unviable or unsupportable fetus and the impacts of violence and demands upon pregnant bodies have revealed that the notion of choice as a *sine qua non* of reproductive autonomy is incommensurate with reality. While the end of Constitutional protection has assured that there will be little distinction between the purposes and outcomes of abortion, the problem of "desire" has revealed the necessity of posing this question within a different framework.

Nevertheless, Webster has made a compelling claim that reproductive autonomy reveals a fragility of desire, a "fantasy of life made and destroyed at will [that] authorizes the destruction of an individual's desire–its enigma, its historical exigencies, its force as a choice among a series of unknown and overdetermined constraints" [12] (p. 274). I ask if refusing abortion serves, beyond the attempted reparation of a narcissistic wound cast by the sense of an undesired child, as the consummation of a death-driven impulse to destabilize the notion of potential. In other words, the death drive forecloses possibility, while only potential inheres in the fetal object. Furthermore, it is important to consider that many abortions happen not out of desire but out of sheer spontaneity and radical uncertainty. The materials of the body are not always subordinate to consciousness. The complex fabric of external circumstances, their violence and their demands do not always produce ready mothers and possible children. The notion that "desire" motivates the framework of the beginning of a narrative and of conception is an assertion of mastery against the very structure of the drive that dismantles it.

For Barbara Johnson, the very poetics of abortion are realized in the fact that the choice for an aborting subject is not a choice "between violence and non-violence, but between simple violence to a fetus and complex, less determinate violence to an involuntary mother and/or an unwanted child" [14] (p. 33). The poetry of abortion is an address to the lost children who have made the poem possible in their absence. Neither romantically nor epically, a poet who writes of their aborted child feels death in its most visceral fashion, not only in their evacuated body but in the products of conception cast into the toilet bowl, the disposal bin and the back of the clinic. The abortion narrative is riddled with a sense of "sacrificial anxiety" [15], but that the subjects who bear the consequences of that anxiety take it up as a matter of course, an essential condition of narrative mastery, is something that only seems to be demanded of the female, transgender, queer, racialized, poor and

young bodies whose inertia is a contingency for a fetal life. If the fetus denies death in the fact that its very life is eternally deferred, then the realization of its susceptibility in being bound to the otherness of the pregnant subject is, in a sense, the traumatic sense of recognition that Tancred encounters in Clorinda's fatal cry and gush of blood. Without the cry of a newborn or the blood of a terminated pregnancy, that scene of recognition never has to happen. The pregnant body should be willing to assume the inertia that is desirable in the fetus so that the death drive can be sublimated into the investments of futurity in what Lee Edelman calls the "cult of the Child", through which "the lives, the speech, and the freedoms of adults face constant threat of legal curtailment out of deference to imaginary Children whose futures, as if they were permitted to have them except as they consist in the prospect of passing them on to Children of their own, are construed as endangered by the social disease as which queer sexualities register" [16] (p. 19). What is, for Edelman, the disciplining of queer sexuality that is sheer non-reproductive enjoyment of sexual climax, the abortion restriction disciplines not merely sexuality but the notion that a living body manifests a narrative after its own interpretive fashion. That abortion means something to each individual subject, whether as an outcome of desire or as a sudden and bloody realization of the volatile and "daemonic" existence at the middle of the intersecting constructs and essences inhabited by any marginalized body, is the main denial of the *Dobbs* decision. Death inheres in a finality of interpretation, and the pregnant subject will only produce a typified narrative of silent, contained fetal possibility.

## 5. Maternal Ambivalence and Abortion as Allegory

In "Sabat Mater," Julia Kristeva describes the worship of the Virgin Mary as the work of "maternal humility." The holy mother of the Christian faith gives life unto death, a child born so he may be sacrificed for the sins of humanity, and she herself is in turn denied death. Mary arrived in heaven through the Assumption so that she did not have to endure bodily profanity. Thus, the repressed mother goddess was brought into unity with the child god of the Christian symbolic order, and motherhood was consecrated as the sublime possibility of death. In Kristeva's words,

> Man overcomes the unthinkable of death by postulating maternal love in its place–in the place and stead of death and thought. This love, of which divine love is merely a not always convincing derivation, psychologically is perhaps a recall, on the near side of early identifications, of the primal shelter that insured the survival of the newborn. Such a love is in fact, logically speaking, a surge of anguish at the very moment when the identity of thought and living body collapses. [17] (p. 252)

The allegorical form is of thinkability, of transforming abundant and abstract concepts into the concrete matter of distinct objects, while Johnson's abortion poetry is predicated on the device of the apostrophe: an address to an absent or inanimate object. This poetic form is rendered as animating in Johnson's work: hence, the bind of an abortion poem manifesting as a call to unrealized children. An allegory presupposes the reader's return to the undifferentiated landscape of the newborn's perception. Objects are not bound or contained but float in the oceanic and eternal mire of the mother–child dyad. In the adult imagination, it is an artifice of simplicity, an outcome of what Jamieson Webster considers to be the desire "on the other side of a divide" between the unlived life of the fetus and the narcissistic fantasy of the adult subject to see the rejecting mother be punished for withholding love from the child. Essentially, the mothers that *Dobbs* confines are not the holy mothers that save death through the redemption of the child but who impose death upon a fetus that, from the adult perspective, has not even known life. That fetal protection legislation so often employs the vocabulary of the "unborn" or "pre-born" reveals, in a sense, the central paradox: that a fetus has life, and therefore can die, but has yet to undertake the defining experience of being a living subject. Death precedes birth in an affront to the human order—not unnatural, but uncivilized.

That I should take up David Cronenberg's film *Crimes of the Future* (2022) as illustrative of the contemporary abortion allegory at least partially owes to the coincidence of its world release with the overturning of *Roe v. Wade.* The film itself is more of a story of evolution gone awry in the slow death of climate change, while the central theme concerns the demands upon an artist to make sense out of disorder and calamity. The story mainly follows performance artist Saul Tenser (Viggo Mortensen), a man with Advanced Evolution Syndrome, whose body spontaneously produces novel organs at an accelerated rate. As humans in this world have lost the capacity for pain and infection, surgery only has glamor in the spheres of art and sex, so Tenser can turn his undisciplined body into art. The extraction and display of his organs in accordance with new bureaucratic regulations for the maintenance of the human form becomes, through Tenser's spectacle, the work of self-mastery. Meanwhile, an underground revolution foments among those who believe that the chaos of the new human body is evolution's answer to anthropogenic climate change, manifesting in a new digestive system capable of consuming plastic. Tenser is employed as an undercover agent investigating these rogue factions while also assenting to the live autopsy of Brecken, the first child to be born with a body made to ingest plastic. It is around Brecken, an eight-year-old corpse, that the plot coalesces. In the film's opening scenes, he is murdered by his mother. His father, the vanguard of the bodily revolution, is the one who asks Tenser to make a spectacle out of his murdered child's body. In Cronenberg's future, the holy child in which the human body is that written in the constitution of the human environment is an object of maternal abjection. The boy's mother, Djuna (Lihi Kornowski), is unable to shake her identification with her son's interiority. While his father must artificially produce the plastic digestive system within himself through surgical intervention, Djuna's focus is the deep embedded memory of having held within herself this apotheosis of an inhuman world:

Djuna: He had this weird, thick white drool that he sometimes slurped all over everything. It was like acid. It could dissolve any kind of plasticky stuff. It would sting if you got it on your skin. It didn't bother him at all. That lizard.
Tenser: Your husband is Lang Dotrice?
Djuna: Was. Was my husband. I disowned him. Fuck him.
Tenser: He invented your son?
Djuna: Yeah. That's how I think of it. And here's another thing I think of. The thought of that slimy worm growing in me still makes me sick. [18] (00:50:39–00:51:26)

Reproductive and sexual terror is a potent theme in Cronenberg's cinema. His body horror consists of grotesque pseudo-genitalia emerging in unexpected places, of desire and destruction coalescing around new points of entry and expulsion as bodily integrity is disintegrated. Cronenberg's horror is often predicated on the susceptibility of the subject, often with an implacable curiosity, to what is unstable, corruptible and destructive in the new cultural milieu. In her book on the death drive in literature and culture, Teresa de Lauretis used no fewer than three Cronenberg films as illustrative of the drive within various vicissitudes. Cronenberg wrote the Freudian nightmare *par excellence* in its confluences of sexual desire and the compromised body. What is particular to this film is not only its situation in a future where putridity is simply a condition of life, but that the aberrant body is the natural one. The film ends with Tenser deciding to retain his novel organs rather than expel them in accordance with the new bureaucratic offices regulating human evolution. What is at stake is not the individual body of a man merging with a fly, a television or a video game but the very constitution of humanity as an exclusive category. The human becomes human because of those evolutionary roads not taken: those rejected and extinguished parts that have dissolved into the primordial mire of pre-history.

In the Freudian domain, nothing past is truly past, as *Beyond the Pleasure Principle* makes very clear in its contentions with evolutionary history. For Freud, the notion that death exists as a condition of organic life (and is, indeed, its primary form) produces yet another irresolvable question of how death originates among living substances. Is it, as the cellular biology of the early twentieth century suggests, a fact that death is an outcome

of multicellularity and sexual reproduction, or is it simply that more complex structures make death visible, giving it a "morphological expression," which cannot be apprehended in unicellular forms? [3] (p. 49). Ultimately, Freud sidesteps the debate of death's origin within a biologistic framework. That death should elude empirical observation at the most singular, basic level of life is a question of theory that subtends Freud's mythopoetic turn. The representative crisis of a death unobserved, a death without meaningful organic expression, is a problem that at once invites and defies witness. Who must prove their death with a cry, and who must hear it in order for there to be a sense of meaning? Additionally, is the creation of meaning around death, mediating it through the contours of an expressive and identifiable body, an act of violence in itself?

The crime of the future is set by the following terms: that fathers are identified with revolution and mothers with termination, or, in the words of Barbara Johnson, that "it is as though male writing were by nature procreative, while female writing is somehow by nature infanticidal" [14] (p. 38). Tenser's novel organs are clearly inscribed in the literary lineage of a fantasized male pregnancy: his assistant, Caprice (Léa Seydoux), describes his developing forms in the manner of an ultrasound technician; he places his hand against the sides where the organs are burgeoning, like an expectant parent; he gazes upon his extracted organ in its display jar at a post-performance cocktail hour with the pride of a newly delivered parent. The film ends with Tenser resolving to keep his baby, to become this new step in human evolution capable of consuming the waste of civilizations. The final shot is the beatific look of the Madonna, who has chosen life through the incorporation, not the rejection, of the abject. The women in the film, even the well-intentioned Caprice, defy this incorporating subjectivity. For them, radical susceptibility is the legacy of infection. For all that is the stuff of the past of the film, the suggestion remains that women are proximate to bodily memory while men are poised to undertake radical new being. The aborting women and pregnant men are the imprints of abortion's death-driven discourse; in the absence of choice or possibility, the women assume a legacy of bodily confinement from which men become liberated. The crime of the future, in a sense, is the crime of an unrelenting grip upon the past. For Djuna, who cannot forget what it was to share the body of the plastic-consuming child, the wasteland of the future can become a new edict of motherly sacrifice.

## 6. Conclusions

The overturning of *Roe v. Wade* is a point of crisis in theory. The lived and inhabited consequences of the decision among pregnant bodies and their eventuated children are beyond the domain of speculation. The death drive is the orienting logic of failure: psychoanalysis's own obsolescence in the violent subjugation of disciplining bodies. The politics of susceptibility enjoin the notion of deliberate consciousness to the uncanniness of indeterminate cause and effect. For those whose bodies are called upon to illustrate the drive against its quiet pulsion and unconscious orientation—the bodies disciplined by the word of dead men and its containment in the halls of authority—the question of how the past will be remembered and the future will be articulated is the catachresis of belief meeting body. In the abundance of abortion, there will always be a voice left unheard, a cry misinterpreted or a shock encounter with an unrecognized subject that so often might be loved and desired. The shock of the Supreme Court's decision is that of a narrative form that takes a new detour toward an ending. How, or in what voice, that ending emerges is in the place of the new storyteller: the post-Roe pregnant subject who cannot terminate the wastes so that man's ambitions might become the artful *mat*(t)*er* of a new, incorporating humanity.

**Funding:** This research received no external funding.

**Institutional Review Board Statement:** Not applicable.

**Informed Consent Statement:** Not applicable.

**Data Availability Statement:** Not applicable.

**Conflicts of Interest:** The author declares no conflict of interest.

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
