# Peer review of "Beyond Choice: Reading Sigmund Freud at the End of Roe"

_philosophies, doi:10.3390/philosophies8060100_

Round 1

Reviewer 1 Report

Comments and Suggestions for Authors

The paper is brilliant, and it was a pleasure to read it.

Comments on the Quality of English Language

I noticed a few minor grammatical errors that a proofreading should catch.

Author Response

Thank you for taking the time to review my manuscript, and for your exceptionally kind comments. I have made revisions according to my feedback from other reviewers, including narrowing down the overall argument and elaborating my claim more clearly and thoroughly. I hope that my revised paper is an improvement on my previous work.  

Reviewer 2 Report

Comments and Suggestions for Authors

This is a complex article with a novel perspective on how Freudian theories and concepts might contribute to a re-reading of the discourse on abortion and fetal personhood and the politics of susceptibility after the overturning of Roe v. Wade. However, some reorganization would be beneficial to strengthen the arguments and facilitate the reader’s comprehension.

Knowledge gap, hypothesis, methods

-The opening statement of the introduction is rather debatable, google scholar yields at least 5 (Remeikis, Pines, Snider, Beck, Webster) more recent articles (than 1969 or 1985) about psychoanalytic literature on abortion. Maybe the knowledge/research gap the article seeks to bridge should be narrowed down.

-The hypothesis „examining the anxieties and impulses bound with“ this new ruling should be reformulated in order to reflect the ambivalences that were observed (and implied in the lines before it is stated what the author will do).

-Also, it should be made clear how the selected primary texts specifically contribute to the overall argumentation. So far, the selection seems rather random.

-What exactly is the methodology used in this article? Is it a deconstructionist reading of the implications of abortion discourses intersecting with Freudian concepts in the primary texts, or is it a discursive analysis of diverging comprehensions of abortion and political susceptibility, or something else.

Structure, voice and language

Despite the skillful engagement with secondary texts that is reflected in the order of the article‘s sections, the overall analysis would be strengthened by a more organic arrangement of the different parts. Otherwise, it seems like an arbitrary sequence of isolated texts with starkly differing lengths.

Concepts like gender, narrative, image appear rather suddenly and might need some contextualization and embedding.

There are many instances when the reader cannot distinguish between the author‘s voice and a secondary source / paraphrase (e.g., on page 4 „reminiscent of a heartbeat“, on page 5 „pro-life terrorism“).

The language is, thus, sometimes a bit harsh, judgmental and sarcastic and might easily be read as biased.

Minor fixes

-differentiation between discourses on a) abortion as an invasive surgical procedure administered on a body, b) abortion as just another liberating way of contraception, or c) abortion as the termination of fetal life – there are many instances when it seems not clear what aspect (or how many of them) are addressed

-perhaps including the information that abortion wasn't a constitutional right before 1973 (especially with the Comstock laws implemented in 1873)

-missing sources, e.g. the numbers on mortality rates on page 3

Comments on the Quality of English Language

There are minor flaws in the usage of idiomatic English, e.g. missing or improper prepositions.

Author Response

Thank you for taking the time to review my manuscript and give me such thorough, constructive feedback. I have taken your recommendations into consideration in rewriting my paper. Please see the attachment for a point-by-point response to your comments. 

Reviewer 3 Report

Comments and Suggestions for Authors

Let me start by identifying myself as a film scholar with an interest in psychoanalysis. The article under review is submitted to the journal Philosophies, and my main criticism concerns, I guess, a certain discrepancy between the discourse of film studies and philosophy. I was intrigued by the topic of reading the discussion on abortion through the lens of Freud's death drive, but I found the argument quite hard to follow from page 6 onwards. For me, it should have been made more explicit what is the function of Johnson's 'poetics of reading abortion'. And why is Zoë Sofia's reading of deathlessness in 2001 relevant? Was Cronenberg's Crimes of the Future aimed at explaining the problem of maternal ambivalence? And quite suddenly Lady Macbeth is introduced, but to what purpose? I lost the train of thought in this latter part, and this can be due to the fact that I am not that well-versed in this discourse of philosophy but I would advise the author to rewrite the latter part by clearly explaining the steps the author is taking and to bear in mind that such steps need to help the reader understand why the upcoming text or film example is relevant for the argument? In short, it was all too implicit and/or too brief for me - which is a pity, for let me say once again, the general approach is interesting. Line 254: note there is a sudden transition to Barbara Johnson. Later, there is a sudden transition to the Star Child Image and to 'complete nuclear annihilation'. Line 381: there is a sudden transition to Lady Macbeth. 

Author Response

Thank you for taking the time to review my paper and provide such helpful feedback. As mentioned in my longer comment (attached below), my argument is really centered upon interpretation and the production of narrative. I hope that, in narrowing down my objects, I have produced a more readable and cohesive paper. 

Round 2

Reviewer 2 Report

Comments and Suggestions for Authors

The author has thoroughly revised the text, and its methodological, structural, and analytical weaknesses have been fixed and corrected. Before it gets line-edited and published, I would suggest expanding the methodology section a bit more in order to flesh out the approach and clarify the method even better (since not every reader might be familiar with the topic). I really enjoyed reading the revised article.

Comments on the Quality of English Language

Has improved considerably, almost flawless academic English.